# Towards a Rapid-Turnaround Low-Depth Unbiased Metagenomics Sequencing Workflow on the Illumina Platforms

**DOI:** 10.3390/bioengineering10050520

**Published:** 2023-04-25

**Authors:** Winston Lian Chye Koh, Si En Poh, Chun Kiat Lee, Tim Hon Man Chan, Gabriel Yan, Kiat Whye Kong, Lalita Lau, Wai Yip Thomas Lee, Clark Cheng, Shawn Hoon, Yiqi Seow

**Affiliations:** 1Bioinformatic Institute, A*STAR (Agency for Science, Technology and Research), Singapore 138632, Singapore; winston_koh@bii.a-star.edu.sg; 2Institute of Molecular and Cell Biology, A*STAR (Agency for Science, Technology and Research), Singapore 138673, Singapore; poh_si_en@imcb.a-star.edu.sg (S.E.P.); kongkw@imcb.a-star.edu.sg (K.W.K.); lalita_lau@imcb.a-star.edu.sg (L.L.);; 3Department of Laboratory Medicine, National University Hospital, Singapore 119228, Singapore; chun_kiat_lee@nuhs.edu.sg (C.K.L.); hon_man_chan@nuhs.edu.sg (T.H.M.C.); 4Department of Medicine, Yong Loo Lin School of Medicine, National University of Singapore, Singapore 119228, Singapore; gabriel_zherong_yan@nuhs.edu.sg; 5Division of Microbiology, Department of Laboratory Medicine, National University Health System, Singapore 119228, Singapore; 6Paths Diagnostics Pte Limited, Singapore 349317, Singapore; 7Genome Institute of Singapore, A*STAR (Agency for Science, Technology and Research), Singapore 138672, Singapore

**Keywords:** metagenomic sequencing, host depletion, DNA/RNA library preparation, liquid biopsy, infectious disease

## Abstract

Unbiased metagenomic sequencing is conceptually well-suited for first-line diagnosis as all known and unknown infectious entities can be detected, but costs, turnaround time and human background reads in complex biofluids, such as plasma, hinder widespread deployment. Separate preparations of DNA and RNA also increases costs. In this study, we developed a rapid unbiased metagenomics next-generation sequencing (mNGS) workflow with a human background depletion method (HostEL) and a combined DNA/RNA library preparation kit (AmpRE) to address this issue. We enriched and detected bacterial and fungal standards spiked in plasma at physiological levels with low-depth sequencing (<1 million reads) for analytical validation. Clinical validation also showed 93% of plasma samples agreed with the clinical diagnostic test results when the diagnostic qPCR had a Ct < 33. The effect of different sequencing times was evaluated with the 19 h iSeq 100 paired end run, a more clinically palatable simulated iSeq 100 truncated run and the rapid 7 h MiniSeq platform. Our results demonstrate the ability to detect both DNA and RNA pathogens with low-depth sequencing and that iSeq 100 and MiniSeq platforms are compatible with unbiased low-depth metagenomics identification with the HostEL and AmpRE workflow.

## 1. Introduction

Infectious disease is a worldwide health burden, with just the eight major infectious diseases (HIV, malaria, measles, hepatitis, dengue fever, rabies, tuberculosis, and yellow fever) causing more than 156 million life-years lost in 2016 [1]. Thus, early detection and diagnosis is important for timely interventions and treatment. Conventional methods, such as culturing of isolated pathogens and molecular amplification of nucleic acids, are widely used in diagnosis, but these methods require prior knowledge or suspicion of the pathogens [2,3,4,5]. In contrast, unbiased (shotgun) metagenomic next-generation sequencing (mNGS), as opposed to a targeted sequencing approach [6], can not only detect multiple pathogens in a single assay but also discover novel or unexpected pathogens, like SARS-CoV-2 during the recent COVID-19 outbreak [3,7,8,9,10,11,12,13]. Conceptually, a dependable, robust metagenomic sequencing solution is an ideal first-line diagnostic if costs are low, time-to-result is fast, and sensitivity is high.

Although metagenomic sequencing technologies are developing rapidly [14,15,16,17], there are certain limitations. First, typical protocols require multiple days to complete due to the time-consuming host background depletion, library preparation, and data analysis processes [3,4]. Second, as the samples contain high human background, sequencing depth needs to be high to account for the excess of human reads, which significantly contributes to incurred costs [5,18,19]. Although a few solutions exist for human background depletion, selective depletion followed by centrifugation enriches only bacteria DNA [18,20], which limits utility by omitting DNA or RNA viruses. Third, most library preparation kits either recommends DNA or RNA as input, but viruses can be both RNA and DNA-based; thus, the need to deploy separate workflows for DNA and RNA preparation adds manpower time, and costs to comprehensive, unbiased sequencing.

To address these, we developed a workflow consisting of a HostEL (Host Elimination) kit—a human background depletion strategy that also allows for enrichment of viruses; and Amplification-Restriction Endonuclease fragmentation (AmpRE)—a single tube DNA/RNA library preparation. HostEL uses magnetic bead-immobilized nucleases to deplete human background after selective lysis, replacing enzyme deactivation with magnetic pull-down, to enrich both pathogen DNA and RNA. AmpRE is a purely additive process for DNA and RNA input with only two clean-up steps before the library is ready for sequencing. It can reduce the processing time while maintaining pathogen signal when performing shallow sequencing on iSeq 100, which allows more rapid diagnosis of communicable diseases. With the two strategies combined, our method can not only detect bacteria but also fungi and DNA and RNA viruses. We conducted an analytical validation for the workflow using plasma with spiked-in microbes (ZymoBIOMICS Microbial Community Standard) at various concentration to mimic the physiological infection conditions down to 70 cells/µL of plasma [21]. The samples were sequenced on two sequencing platforms, iSeq 100 and MiniSeq, with three different sequencing turnaround time to cater to different clinical workflow needs and demonstrated high sensitivity and specificity with each platform. We also conducted a clinical validation with 42 plasma samples, with 93% of the samples agreeing with the qPCR results on the clinical samples.

## 2. Materials and Methods

### 2.1. Analytical Validation

ZymoBIOMICS Microbial Community Standard (Appendix A), consisting of 8 bacterial and 2 fungal strains at 1.4 × 10^7^ total cells/µL, was diluted 100×, 250×, 1000× and 2000× before spiking 5 µL into 500 µL of human plasma to yield at 1.4 × 10^6^, 5.6 × 10^5^, 1.4 × 10^5^, 7.0 × 10^4^ total cells/mL of plasma, respectively. The spiked-in plasma was either HostEL-depleted (Paths Diagnostics, Singapore) or just bead beaten. For HostEL-treated samples, 30 µL of incubation buffer and 10 µL of nuclease beads were mixed with 500 µL of plasma and incubated for 20 min shaking at 37 °C. The nuclease beads were then attracted with a magnetic rack and supernatant bead-beaten for 30 s before extraction using Zymo Quick DNA/RNA Viral extraction kit (Zymo Research, Irvine, CA, USA), eluting in 30 µL. The control samples were just bead-beaten before extraction. Libraries were prepared with AmpRE kit (Paths Diagnostics) as per manufacturer’s instructions. 7 µL of TNA was then used as input into the AmpRE kit. First strand synthesis in 10 µL reaction is followed by the addition of 10 µL second strand reagent for second strand synthesis, then an addition of 30 µL amplification mix was added for amplification with methylated nucleotides. Cleanup with SPRI beads and elution in 10 µL was followed by digestion for 4 µL of eluted DNA in a 5 µL reaction for restriction digest of methylated nucleotides. An amount of 10 µL of ligation buffer was added to the fragmented DNA for ligation and tailing of sequencing arms before amplification and barcoding. The sample was then cleaned up using SPRI beads and eluted in 10 µL and quantified on Agilent 4150 TapeStation Instrument (Agilent Technologies, Santa Clara, CA, USA) using Agilent D5000 ScreenTape System (Agilent Technologies). Samples were pooled and sequenced on iSeq 100 and MiniSeq (Illumina, San Diego, CA, USA), respectively.

### 2.2. Sequencing on iSeq 100 System and MiniSeq System

Each quantified sample using the Agilent Tapestation was diluted to a concentration of 190 pM with 10 mM Tris HCl, pH 8.5. Up to 5 samples were then pooled together in equal volume, making up to a total of at least 20 µL for sequencing on the iSeq100 system and up to 20 samples were pooled for the Minseq system to ensure that there are sufficient reads per sample across the systems. An amount of 20 µL of the pooled sample was loaded onto the iSeq 100 cartridge before loading onto iSeq 100 for sequencing. For MiniSeq, each quantified sample was diluted to a concentration of 10 nM with RSB (10 mM Tris-HCl, pH 8.5 with 0.1% Tween 20) and pooled at equal volume before diluting down to 1 nM with RSB. The pooled library was denatured by mixing 5 µL of 1 nM pooled sample and 5 µL 0.1 N NaOH and incubating at room temperature for 5 min. 5 µL of 200 mM Tris-HCl, pH 7.0, was then added to the denatured library. The denatured library was diluted down to 5 pM by adding 985 µL of pre-chilled Hybridization buffer (HT1) to the 15 µL of denatured library. The library was further diluted down to loading concentration of 1.6 pM by mixing 160 µL of 5 pM denatured libraries with 340 µL of prechilled HT1. 500 µL of the diluted sample was loaded onto the MiniSeq cartridge before putting onto the system for sequencing.

### 2.3. Data Analysis

Fastq sequence files were exported from the sequencers, and quality control was performed using Prinseq tool, specifically to remove low complexity reads, entropy is set to be 0.4, with quality trimming on both left and right side of the reads up to 10 bases [22]. Reads that are less than 20 bp in length after trimming are removed and resulting high quality reads was then aligned to the human genome using HiSAT2 v2.21 using the GRCh38 reference [23]. On average, ~88.4% of reads were mapped to the human genome. These are considered to be human background reads and removed from downstream analysis. There can be the possibility of reads from non-human sources sharing homology with the human genome being removed at this stage. The remaining unaligned reads can thus be considered as unique reads that do not share homology with human background. These remaining reads were then identified using KRAKEN2 v2.1.2 and BRACKEN v2.6.2 tools. For KRAKEN2, the confidence level is set to 0.05 and was specified to report the minimizer data for manual inspection of assigned reads. As KRAKEN is not an alignment based algorithm, when inspecting and visualizing assigned reads, reads assigned to specific microbes are first aligned to their respective NCBI reference genomes using BWA v0.7.17-r1188, then visualized using IGV v11.0.9.1 and R v4.0.2. The KRACKEN2 database used is a custom database, which is a combination of the standard database with Fungi added on downloaded from (https://benlangmead.github.io/aws-indexes/k2 (accessed on 2 March 2020)). At least five BRACKEN assigned reads to the species was the cutoff for a positive detection. For the simulated iSeq 100 run, the reads were trimmed down to 100-bases pair-end reads using AWK before processing through the same pipeline. Statistical analysis of sensitivity, specificity and clinical agreement was then performed in R. For the analysis, we defined these statistics as Sensitivity = True positives/(True positives + False negatives) and Specificity = True negatives/(True negatives + False positives); where true positives are the number of spiked-ins that were correctly identified by sequencing as positive. False positives are the number of spike-ins that were incorrectly identified by the sequencing as present in negative plasma. True negatives are defined as the number of spike-ins that were correctly identified by being not present in negative plasma. False negatives are the number of spike ins that are identified by the sequencing test in negative plasma. For the analysis on clinical agreement, we illustrate the numbers in a cross tabulation illustrating the of occurrence of the different combinations of situations of positive/negative clinical diagnosis vs. sequencing positive/negative.

### 2.4. Sample Collection and Ethics Statement

The studies involving human participants were reviewed and approved by NHG DSRB Study Reference Number: 2017/00632. The patients/participants provided their written informed consent to participate in this study. An amount of 5 mL of blood was collected in EDTA tubes, and plasma was obtained by centrifugation at 2000× *g* for 10 min. 500 µL was aliquoted out and stored at −80 °C before HostEL and AmpRE (Paths Diagnostics) processing as per manufacturer’s instructions. The samples were prepared as for analytical validation but extracted with EZ1 Virus Mini 2.0 kit (Qiagen, Hilden, Germany) instead, eluting in 60 µL of water.

### 2.5. qPCR Validation

An amount of 1 µL of extracted samples and 0.3 µM of each primer for pathogen (Appendix A) was added to each 20 µL reaction. For bacteria, fungi, and DNA viruses, a Maxima SYBR Green/ROX qPCR Master Mix (Thermo Fisher Scientific, Waltham, MA, USA) was used. For RNA virus, Toyobo Thunderbird One Step qRT-PCR kit (Toyobo, Osaka, Japan) was used with 1× SyBr Green (Thermo Fisher Scientific). The mixture was amplified on the QuantStudio 1 qPCR system (Thermo Fisher Scientific) using the thermal profile: 50 °C for 15 min, 95 °C for 2 min for the initial denaturation, followed by 60 cycles of 95 °C denaturation (15 s) and 60 °C annealing and extension (60 s). Melting curve analysis was performed by ramping the temperature up to 95 °C after qPCR to verify the specificity and identity of each of the PCR product. The list of primers used is shown in Appendix A.

## 3. Results

### 3.1. Principles of HostEL and AmpRE

The HostEL kit relies on nucleases were conjugated onto magnetic beads and mixed into 500 µL plasma samples and 30 µL of a Tris-HCl pH 7.5 buffered solution containing 8.25% *w*/*v* CHAPS hydrate detergent to lyse lipid membranes to release protected nucleic acids, similar to other host depletion methods [24]. The samples were then mixed for 30 min at 37 °C before the beads were removed. As the nucleases are conjugated onto beads, these are effectively removed prior to extraction and cannot then result in degradation of signal. This is particularly important for RNase as any carryover can be detrimental to metagenomic RNA signal (Figure 1a).

AmpRE is a library preparation kit that uses random primers for both first and second strand synthesis. Once the double stranded library is made, methylated cytosines are incorporated at random through PCR such that the median fragment size derived after cutting with a methylation-dependent restriction endonuclease results in a median of around 400 bp. SPRI purification is performed after PCR, followed by enzymatic fragmentation before universal primers are ligated on and the amplicons amplified by PCR. Barcoding primers are added last before SPRI purification to remove background primers (Figure 1b). Through empirical optimization of the protocol, it can take both DNA and RNA as input and assay both DNA and RNA pathogens.

### 3.2. Low Depth Sequencing Can Detect Physiological Levels of Bacteria

The mNGS workflow utilized includes a host depletion stage with HostEL (Figure 2A), followed by bead-beating and total nucleic acid (TNA) extraction. Sequencing libraries are then generated with a 4 h additive library preparation protocol before sequencing in batches of 5 samples on iSeq 100, a sequencing platform which can produce ~4 million reads in 19 h, to achieve low depth <1 million reads per sample. The iSeq 100 was selected for the proof-of-principle as it utilizes a plug-and-play cartridge that is user-friendly for the clinical laboratory. The reads were aligned to the human genome to discard background before identifying pathogen sequences and quantifying read counts in under 2 h (Figure 2B).

To validate our workflow (Figure 3A), human plasma was spiked with various dilutions of ZymoBIOMICS Microbial Community Standard, which comprises of bacteria and fungi (Appendix A), to achieve the total cell counts of 1.4 million, 560,000, 140,000, and 70,000 cells per mL of plasma, respectively. This mimics the physiological concentrations of 10^5^ copies/mL in the body during active infection [21]. Spike-in samples without HostEL treatment and samples with no spike-in were prepared as negative controls for comparison, which made up a total of 10 samples. Even with a median of 56,498 paired end reads (2 × 150 bp) per sample across all samples on the iSeq 100, the pipeline identified all 10 species at all dilutions when treated with HostEL while preserving relative distribution of species. In contrast, the untreated samples started losing signal at 140,000 and 70,000 cells per mL (Figure 3B). The enrichment is ~2-fold at 1.4 million cells/mL of plasma of HostEL treated samples and extend to ~4-fold at the lower concentrations on iSeq 100, while the enrichment on MiniSeq is at least 1.5-fold (Appendix A). In this study, the lowest limit of detection (LOD) was set to 70,000 cells per mL of plasma; the equivalent of a total of 35,000 cells in 500 µL of plasma (Figure 3C), demonstrating the depletion of background host reads. The relative distribution of species was preserved at all concentrations, suggesting low-depth (<1 million reads per sample) sequencing allows for detection of microbial species in a high-human-background matrix.

### 3.3. HostEL Enhances Sensitivity and Specificity of Low Depth Sequencing

To establish analytical sensitivity and specificity of our approach, we prepared and sequenced 11 ‘blank’ plasma samples and 4 samples each at the above-mentioned dilutions with and without HostEL treatment. Using a stringency metric that is based on the minimum number of read counts to call the presence of an organism to reduce the chance of false positive, we evaluated the sensitivity and specificity of the platform at detecting all 10 spiked-in species (10 species per sample). With HostEL treatment, we achieved a sensitivity of 96% and a specificity of 96%, while the values for the control were 82% and 97%, respectively (Figure 4A). Further breakdown of the sensitivity at different dilutions demonstrate the effect HostEL treatment has on the sensitivity of detection, with the lowest concentration registering 88% against 63% of the control sample (Figure 4B). In conclusion, the mNGS workflow shown here enables the detection of bacterial and fungal pathogens with low-depth sequencing.

### 3.4. Concordance with Banked Patient Plasma Samples Is >90%

Clinical validation of our mNGS workflow was conducted on 42 patient plasma samples collected in compliance to IRB guidelines. These samples have previously been clinically diagnosed with nucleic acid tests to be positive (34 samples) or negative (8 samples) for pathogen infections (Table 1). To understand the correlation between sequencing workflow and the Ct values, and the LOD of the sequencing platforms, qPCR with targeted primers was conducted on 8 Hepatitis B samples diagnosed clinically with qPCR (Appendix A). Correlation plots were plotted with Ct values against the sequencing reads, and linear correlation was observed (Figure 5A). As we are targeting 100–500 k reads per sample, we can only confidently detect microbial sequences with an abundance of 1 in every 10 k reads. For samples with Ct~30, the relative abundance is about 1 in 10 k reads; thus, we have high confidence of our sensitivity up to a corresponding Ct value of 30 and below (Appendix A). When qPCR validation was also completed on all sequence-negative clinical samples as quality controls (Table 1) and some sequencing-positive samples as positive controls. qPCR was unable to detect any pathogen in 3 out of 13 of these samples, while the rest had a Ct ≥ 33. The high Ct values for these samples may be due to the following reasons: (i) pathogens may be present in a low level, and (ii) sample integrity may have been impacted, or samples had degraded due to long-term storage.

We further compare the percentage agreement of our workflow with the positive clinical molecular diagnosis data and after excluding the 13 samples that were deemed to be low quality (Ct ≥ 33 or undetectable) from the statistics (Figure 5B). We obtained 93% agreement with the clinical results. To ensure that there is no false positive in our workflow, 8 control plasma samples (B068–B075) clinical negative for pathogen were also analyzed with the workflow (Table 1). Our workflow shows 93% agreement on iSeq 100 on both positive and negative samples with Ct < 33 (Figure 5B). In rare cases, with Ct ≥ 33 where sample quality is low (B048), pathogen detection is still possible, although with a lower sensitivity (Table 1).

### 3.5. Both DNA and RNA Viruses Can Be Detected If Ct Is Less than 33

Both DNA viruses (Hepatitis B, CMV, and EBV) and RNA viruses (Hepatitis C, Dengue, HIV) were detected from the clinical samples (Table 1), suggesting that AmpRE library preparation kit enables the detection of both DNA and RNA pathogens. Unlike the analytical validation when the extracted TNA was eluted in 30 µL, the clinical samples were eluted in 60 µL of water based on clinical SOP. With only 7 µL used as input into the library preparation, we were effectively sampling 58 µL of plasma (500 µL × 7 µL/60 µL) to detect the pathogen. With a more aggressive concentration strategy during nucleic acid extraction, the sensitivity will likely increase.

### 3.6. Rapid Turnaround Can Be Achieved with Different Sequencing Platforms

Our workflow was tested on the iSeq 100 2 × 150 sequencing platform due to its ease of use, which would be a big factor in clinical deployment of a metagenomic sequencing solution. However, if implemented in its entirety, the whole workflow will deliver the results to the infectious disease specialist in 27 h (Figure 2C). In theory, with a sample submission cutoff at 11 am and samples loading at 5 pm, the report will be out at 2 pm the next day with our workflow. If the sequencing time was reduced to 13 h, then it is possible to deliver results by 8 am the next morning forward rounds. To simulate such a workflow, we trimmed the reads to 2 × 100 paired end to see if that affected the sensitivity and specificity of such a low-depth (<1 million reads per sample) approach to metagenomic sequencing (‘iSeq-trimmed’). In addition, we evaluated the MiniSeq, which can deliver a single end 100 nucleotide read in 5 h, potentially shortening the entire workflow to 13 h, albeit with the trade-off of greater user involvement in setting up the MiniSeq run. To compare the low-depth sequencing, the read count was kept at 1 million reads per sample (MiniSeq: 20 samples per sequencing run; iSeq100: 5 samples per sequencing run).

The analytical validation using 4 (iSeq-trimmed) or 1 sample (MiniSeq) at each dilution level with negative controls obtained comparable sensitivity with the 19 h protocol for both the trimmed and MiniSeq run (Figure 6A). The MiniSeq, in particular, resulted in a higher sensitivity without HostEL treatment than the iSeq 100, albeit with a much smaller sample size. Although the sample size was small, it suggests the specificity is comparable for all 3 protocols (Appendix A). Thus, we can conclude that reducing the run time with shorter protocols seems to result in no loss of sensitivity.

Correlation plots of Ct values against the sequencing reads were also plotted for the MiniSeq, and linear correlation was also observed (Figure 6B, Appendix A). With the patient samples and a Ct 33 cutoff, MiniSeq performed equivalently to the iSeq 100 run (Figure 6C). Our results suggest that the HostEL and AmpRE workflow is compatible with all three sequencing setup and can deliver rapid unbiased metagenomic sequencing results with shallow sequencing.

## 4. Discussion

Metagenomic next-generation sequencing (mNGS) has been used extensively in the detection of infectious diseases and is found to have certain advantages over conventional methods. It is an unbiased approach that can detect both known and unknown pathogens. Huge amount of effort has been made in key steps of mNGS, such as human background depletion, non-human sequence enrichment, library preparation, etc., to improve the diagnosis efficiency [8,25,26,27,28,29,30,31,32]. Nevertheless, challenges, such as cost and simplicity of mNGS method, still exist [3]. In this study, we addressed these challenges by establishing a workflow that uses HostEL and AmpRE kits for human background depletion and library preparation.

There are a few challenges involving in human background depletion. First, the soluble nucleases are hard to remove from the reactions. Second, it is hard to fine tune the preferential lysis. We developed the strategy, HostEL, for preferential lysis of human cells followed by enzymatic digestion of human background using enzyme-immobilized magnetic beads, leaving intact pathogens for total nucleic acids (TNA) extraction and sequencing. As only a single step of magnetic beads removal is needed to remove the enzymes without the need for enzyme deactivation, it can be applied to biofluids of as low as 500 μL. This strategy allows easy integration into any clinical system as it is a simpler and shorter process. We validated HostEL in our analytical validation using ZymoBIOMICS Microbial Community Standard, consisting of 8 bacteria and 2 yeasts, as a spike-in control to mimic the physiological infection condition. As illustrated in our study, HostEL is able to deplete human nucleic acid without compromising on the composition of the pathogens. At 1.4 million cells/mL of plasma, HostEL is able to increase non-human reads by ~4-fold. When the microbial contribution was diluted to 70,000 cells/mL of plasma, a similar level to the physiological level of 10^5^ copies/mL as reported in Darton et al., 2009 (30), the composition of the pathogens remained consistent. This appears to be superior to other depletion methodologies available in terms of maintaining relative composition [33]. In a separate study, non-human fraction of the sequencing reads in mNGS was found to be as low as less than 0.1% of the total reads [34]. Hence, with the use of HostEL, we are able to save cost on mNGS as the same amount of data is obtainable from less sequencing data. There is a question on the limit of detection and sensitivity of this approach as opposed to more sensitive qPCR, targeted sequencing approaches or even higher depth unbiased metagenomic sequencing, especially with the Ct33 detection cutoff for the banked clinical samples. While health economics calculations are not beyond the scope of this article, the availability of a first-line diagnostic for febrile neutropenia or fever of unknown origins that can detect an active infection of viruses, bacteria, or fungi at a comparatively low price point can save the patient weeks of diagnostic workup and provide timely targeted therapeutic intervention. This has the potential to reduce hospitalization duration and can be beneficial for antibiotic stewardship by only targeting the right organisms, potentially at the expense of missing residual infections that are not detectable in the blood.

Typical strategy for sequencing library construction for DNA and RNA is to have two separate library preparations. The AmpRE kit is to combine both library preparations into a single pot, which reduces the cost since the same reagents are used. As the kit is fast (under 4 h) and can be used for input as low as 0.7 ng of TNA, we are able to obtain adequate signal from 500 μL of biofluids after treatment with HostEL. Sequencing of AmpRE generated libraries can be completed on demand in small batches on iSeq 100, allowing rapid detection of both DNA and RNA viruses, bacteria, and fungus if sufficient samples are obtained daily. At 5 samples per run, we believe the sample load in a tertiary hospital should enable daily sequencing runs with the corresponding rapid speed of result delivery. Currently, AmpRE is optimized to produce libraries of certain sizes that are better for short-read sequencing. However, the strategy can be tuned for other sequencing platforms. Unlike most library preparation kits, AmpRE is also a pure additive process with minimal clean-up steps, thus reducing the time and effort needed. Nonetheless, there is risk of having background amplicons in the samples, and hence our workflow is optimized to prevent such occurrence.

In our clinical validation cohort of 42 plasma samples, which consist of DNA and RNA viruses and bacteria (Table 1), HostEL and AmpRE workflows are able to achieve high agreement with the clinical molecular diagnosis data. We performed the sequencing on two separate sequencing platforms, iSeq 100 and MiniSeq, and a simulated short iSeq 100 run and achieved concordance results of 90% agreement with the positive clinical molecular diagnosis samples. Our workflow also showed 100% agreement with the negative clinical molecular diagnosis samples, ruling out the high possibility of having false positive result. We further demonstrated that sequencing reads scale linearly with the pathogen load by conducting qPCR on clinically diagnosed Hepatitis B samples. Correlation plots of the Ct value against sequencing reads show that sequencing reads increase linearly with the decreasing Ct values, but the limit of detection of our sequencing platforms dropped when Ct > 30. Our initial results of our workflow are promising; however, they are limited to small diversity of pathogens and biofluid due to logistical limitation and timeline. We intend to extend our workflow to larger diversity of pathogens with different specimen types. We acknowledge that there are limitations to our workflow as it is not applicable to parasite infections, as parasites will not survive under selective lysis assay.

We also validated this workflow on iSeq 100 and MiniSeq to demonstrate the portability of the assay between different Illumina platforms at different sequencing depths and different run times. While sequencing workflows can theoretically be performed in under 24 h in ideal conditions, there are real-world limitations, such as operational hours of diagnostic laboratories, requirement for batching to minimize sequencing costs, analytical pipeline timing is frequently omitted from the calculations and cost considerations (3). To enable unbiased metagenomic sequencing to be a first-line diagnostic option, there is a need to deliver cost-competitive solutions that fit clinical laboratory setup. In the 3 workflows presented, the relatively short sample preparation time (6 h) coupled with competitive sequencing costs (Figure 2A), the option of different workflows and rapid sample analysis (2 h) enable hospitals to offer unbiased metagenomic sequencing as a diagnostic tool with next-day turnaround routinely. The option of the simplified iSeq 100 workflow can work well with labs with less skilled technicians at the cost of longer turnaround, while the MiniSeq offers speed at the cost of setup complexity. All three protocols have similar detection capabilities and perform well with clinical samples. Thus, we believe the workflow presented in this publication can potentially benefit the clinical community.

## 5. Conclusions

In conclusion, we established a mNGS workflow using HostEL and AmpRE, which considerably reduces time, cost, and effort by effectively depleting human background and amplifying both DNA and RNA into sequencing library in a single reaction, as illustrated in our analytical and clinical validations. We plan to explore fungal infection and also infections in a diversity of biofluids and anticipate to scale up the number of clinical samples in our validation to further support the efficiency of our strategies in different clinical settings.

## 6. Patents

Patents have been filed for the HostEL and AmpRE workflows.

## Figures and Tables

**Figure 1 bioengineering-10-00520-f001:**
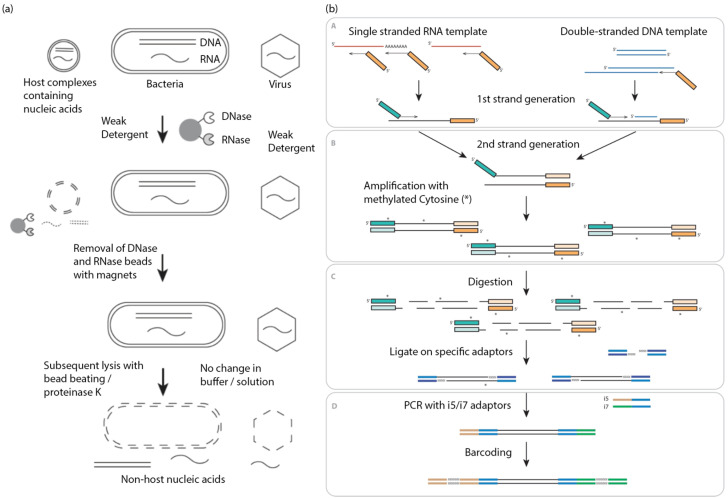
HostEL and AmpRE workflows (**a**) HostEL makes use of selective lysis with nucleases conjugated on microbeads that can be removed magnetically to remove nuclease from downstream reactions without the need for centrifugation, allowing the retention of viral particles; (**b**) AmpRE allows for simultaneous processing of DNA and RNA in single pot reaction followed by amplification, methylation-based restriction digestion for fragmentation, ligation of sequencing adaptors, and barcoding. * represents the presence of the methylated cytosine in the DNA library.

**Figure 2 bioengineering-10-00520-f002:**
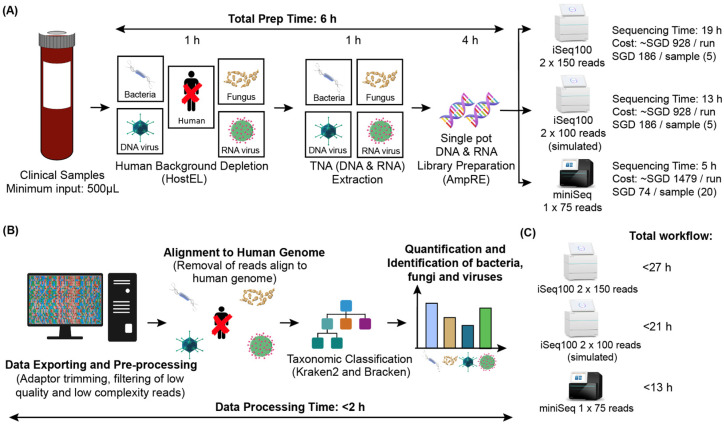
Schematic diagram of mNGS workflow using HostEL and AmpRE. (**A**) Workflow of mNGS using HostEL and AmpRE. A minimum of 500 µL of a clinical sample, such as plasma, is treated with HostEL to deplete human background before total nucleic acids (TNA) are extracted. TNA is then prepared into a sequencing library with AmpRE, followed by sequencing on the iSeq 100 (2 × 150; 2 × 100 simulated) or MiniSeq (1 × 100). The time required for each step and the cost of sequencing are indicated. (**B**) Workflow of data analysis. Sequenced data is exported in Fastq format from the sequencing platforms, and high quality and compatibility reads are selected. These sequences are then aligned to human genome, where the matched human sequences are removed. The reads of the pathogens (bacteria, fungus, and both DNA and RNA viruses) are then quantified and identified using KRAKEN2 and BRACKEN tools. The attached time for each step is included for reference. (**C**) Comparison of the different sequencing platforms on the total workflow time.

**Figure 3 bioengineering-10-00520-f003:**
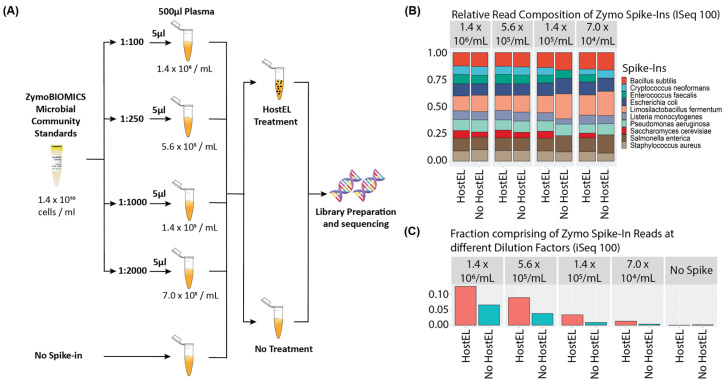
(**A**) Serial dilution of physiologically relevant levels of the ZymoBIOMICS Microbial Community Standard containing 8 bacteria and 2 yeast (Appendix A) to the estimated total microbial cells/mL concentration in plasma is completed before treatment with or without HostEL followed by TNA extraction and AmpRE library preparation. A total of 5 samples are batched and sequenced on the iSeq 100. (**B**) Relative contribution of microbial spike-ins towards non-human reads across different dilution factors of HostEL-treated and control samples mapping to the 10 organisms. The relative ratio of the organisms was preserved in the HostEL treated samples at all dilutions, while the control samples lost some organism reads at higher dilutions. (**C**) Fraction of reads that mapped to spiked-in microbes at different dilutions with or without HostEL treatment. HostEL-treated samples significantly increase the percentage of reads across all different dilution factors as compared to the non-treated samples. Detailed percentages of the reads are shown in Appendix A.

**Figure 4 bioengineering-10-00520-f004:**
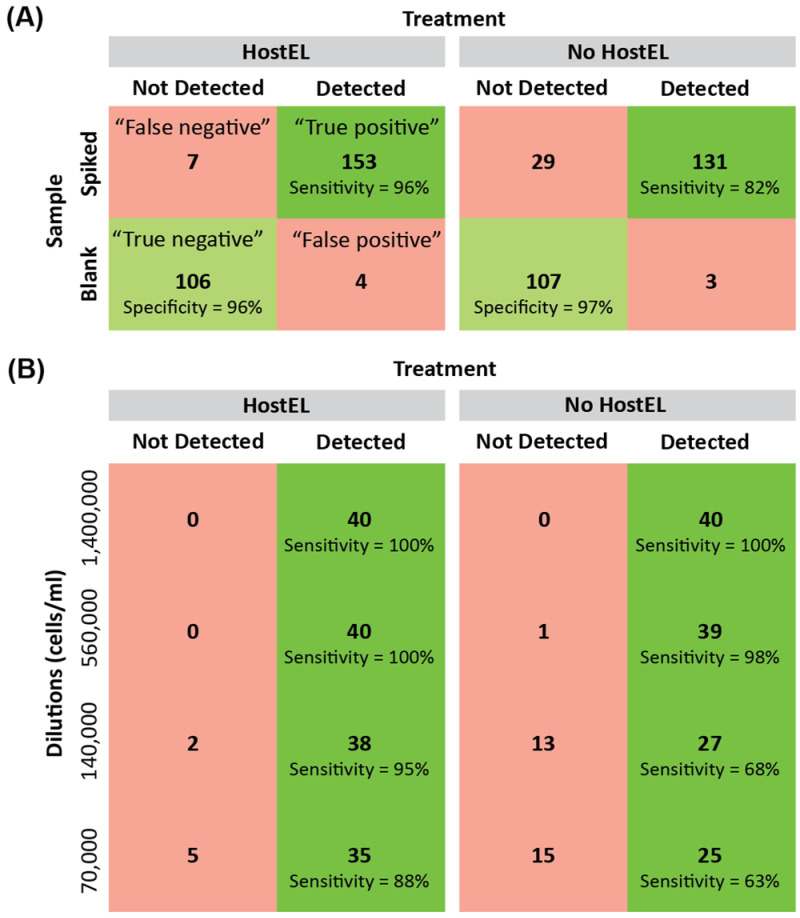
11 blank plasma and 4 repeats of each dilution of the 10 microbial standards (for a total of 40 expected spike-ins to be detected) were prepared and sequenced on the iSeq 100 with a high stringency cutoff to prevent false positives. Based on the 10 organisms spiked into the plasma, a diagnostic truth table was constructed (**A**) with and without HostEL treatment and further broken down into (**B**) detailed sensitivity at different dilutions.

**Figure 5 bioengineering-10-00520-f005:**
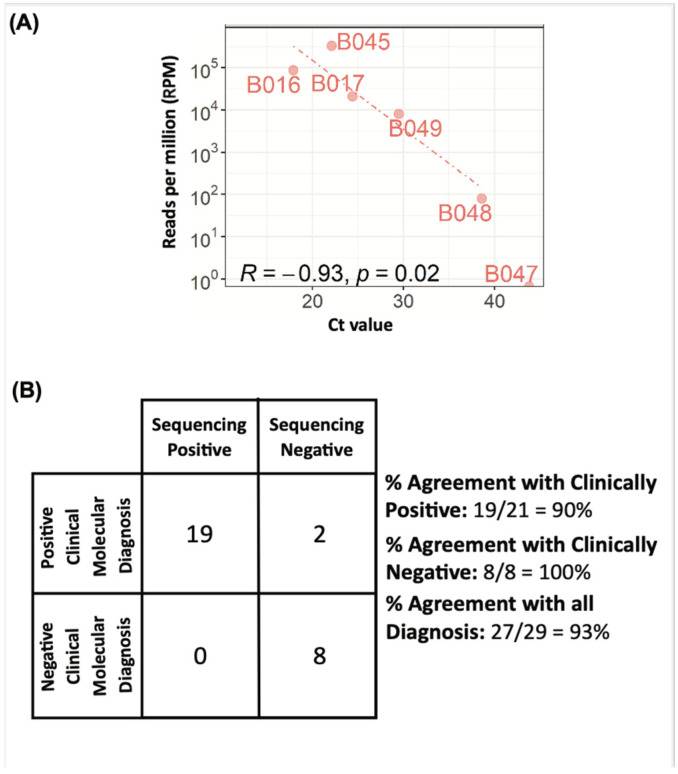
Clinical validation of HostEL and AmpRE workflow. A total of 500 µL each of 42 banked patient plasma samples of infected patients or control plasma was put through the workflow as before, with the EZ1 virus mini 2.0 kit rather than a manual extraction kit. A total of 7 µL was processed by AmpRE and 5 samples were batched and sequenced on iSeq 100. (**A**) Correlation between sequencing reads and qPCR results for Hepatitis B-positive samples (Appendix A). We performed qPCR on 8 diagnosed Hepatitis B samples to check the correlation between sequencing reads and pathogen loads. The Ct values were plotted against the sequencing reads. Sequencing reads of both sequencing platforms correlate linearly with the Ct values. For samples with Ct ≥ 33, both sequencing platforms can only pick up less than 10 per million reads or have stochastic dropout of these pathogens (Table 1). (**B**) Analysis of clinical validation of HostEL and AmpRE workflow on plasma samples. Our cohort showed concordance results of 100% agreement between the workflow for all the positive clinical molecular diagnosis plasma samples below Ct 33. To ensure there is no false positive, we conducted the workflow on 8 negative clinical molecular diagnosed samples and obtained 100% agreement.

**Figure 6 bioengineering-10-00520-f006:**
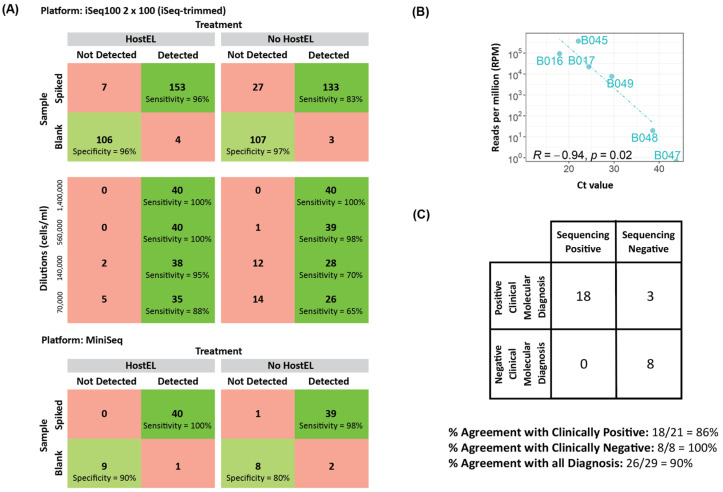
Clinical options for turnaround time. We computationally trimmed the reads of the iSeq 100 to 2 × 100 to simulate a 13 h run time and also ran the same libraries batched with 20 samples per run on the MiniSeq. (**A**) Sensitivity and specificity performance on the spiked-in samples for the trimmed iSeq 100 and MiniSeq runs were similar to the full iSeq 100 run. Breakdown of the different dilution levels for iSeq-trimmed is also included. (**B**) Correlation between sequencing reads and qPCR results for Hepatitis B-positive samples (Appendix A) (**C**) MiniSeq diagnostic truth table for samples with Ct < 33.

**Table 1 bioengineering-10-00520-t001:** List of pathogens detected from clinical diagnosis, iSeq 100, and MiniSeq.

Sample	Clinical Diagnosis	iSeq 100 Detection	iSeq 100 Count	MiniSeq Detection	MiniSeq Count	qPCR Ct
B012	Hepatitis C Virus	Hepatitis C Virus	29	Hepatitis C Virus	72	23.8
B013	Hepatitis C Virus	Hepatitis C Virus	32	Hepatitis C Virus	66	27.2
B014	HIV-1	HIV-1	10	HIV-1	26	-
B015	HIV-1	HIV-1	2	HIV-1	3	-
B016	Hepatitis B Virus	Hepatitis B virus	4238	Hepatitis B virus	11,394	17.9
B017	Hepatitis B Virus	Hepatitis B virus	495	Hepatitis B virus	1226	24.4
B018	Hepatitis E Virus	-	0	-	0	34.3
B019	Hepatitis E Virus	-	0	-	0	37.1
B020	Cytomegalovirus	Cytomegalovirus	9	Cytomegalovirus	10	-
B021	Cytomegalovirus	Cytomegalovirus	22	Cytomegalovirus	100	-
B022	Varicella Zoster	-	0	-	0	33.0
B023	BK virus	-	0	-	0	32.0
B024	BK virus	-	0	-	0	33.0
B025	Epstein–Barr virus	Epstein–Barr virus	13	Epstein–Barr virus	2	-
B040	Dengue type 4	-	0	-	0	Neg
B041	Dengue type 2	Dengue virus	2	Dengue virus	3	-
B042	Dengue type 1	-	0	-	0	Neg
B043	Dengue type 3	Dengue virus	7	Dengue virus	18	-
B045	Hepatitis B Virus	Hepatitis B virus	7427	Hepatitis B virus	23,336	22.1
B046	Hepatitis B Virus	-	0	-	0	Neg
B047	Hepatitis B Virus	-	0	-	0	43.8
B048	Hepatitis B Virus	Hepatitis B virus	2	Hepatitis B virus	2	38.6
B049	Hepatitis B Virus	Hepatitis B virus	118	Hepatitis B virus	318	29.5
B050	Hepatitis B Virus	-	0	-	0	Neg
B051	Hepatitis C Virus	Hepatitis C Virus	17	Hepatitis C Virus	128	23.3
B052	Hepatitis C Virus	Hepatitis C Virus	6	Hepatitis C Virus	8	26.7
B053	Hepatitis C Virus	Hepatitis C Virus	2	Hepatitis C Virus	20	24.8
B054	Hepatitis C Virus	-	0	-	0	35.0
B056	Hepatitis C Virus	Hepatitis C Virus	4	-	0	32.9
B057	Hepatitis C Virus	-	0	-	0	35.9
B058	Hepatitis C Virus	-	0	-	0	34.1
B059	Hepatitis C Virus	Hepatitis C Virus	7	Hepatitis C Virus	2	28.8
B060	Hepatitis C Virus	Hepatitis C Virus	8	Hepatitis C Virus	2	27.9
B061	Hepatitis C Virus	-	0	-	0	30.1
B068	No Infection	-	0	-	0	-
B069	No Infection	-	0	-	0	-
B070	No Infection	-	0	-	0	-
B071	No Infection	-	0	-	0	-
B072	No Infection	-	0	-	0	-
B073	No Infection	-	0	-	0	-
B074	No Infection	-	0	-	0	-
B075	No Infection	-	0	-	0	-

Green: Clinically diagnosed pathogen was detected on iSeq 100/MiniSeq; Red: Clinically diagnosed pathogen was not detected on iSeq 100/MiniSeq; Blue: Ct value < 30; Yellow: Ct value > 30 or negative (undetectable).

## Data Availability

The analytical validation sequencing dataset for this study can be found in the figshare repository https://doi.org/10.6084/m9.figshare.21694184. The clinical validation dataset contains patient genomic data and is available for download upon request, pending ethics approval for each individual request.

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
