# Peer review of "Towards a Rapid-Turnaround Low-Depth Unbiased Metagenomics Sequencing Workflow on the Illumina Platforms"

_bioengineering, 2023, doi:10.3390/bioengineering10050520_

Round 1

Reviewer 1 Report

The author established the HostEl and AmpRE workflow to improve the pathogen clinical diagnosis. Compared to current techniques, the method mentioned in the paper tends to be more sensitive and faster. However, there are still some points not clearly explained.

1.      In HostEL part, how to make sure the human background elimination is efficient?

2.  In the data analysis part, the author mentioned the analysis will remove the human-matched sequences. However, there are some highly conserved sequences between humans and pathogens, is there any way to confirm that the sequences removed are only human background?

3.      In ZymoBIOMICS Microbial Community Standard test, the author uses 8 bacteria and 2 fungal genes to test the method, and in the patient practice, why only include virus tests but not bacteria and fungi? Compared to the ZymoBIOMICS Microbial Community Standard test, the clinical samples were more complicated and meaningful. So I think it is better to include bacteria-infected and fungi-infected samples to support the previous result.

There are some typo and mistakes in text and figures.

1. Please revise Fig 1 panel B, words need to be reversed.

2.   In Line 166, RNAse should be RNase.

Reviewer 2 Report

Infectious disease is a worldwide health burden causing millions of life-lost each year. Early detection and diagnosis are critical for timely interventions and treatment. The traditional methods usually require prior knowledge and can only detected several suspicious pathogens in a single assay. In this study, the authors established an efficient unbiased metagenomic sequencing workflow in which the human background was depleted first and both DNA and RNA pathogens can be detected. The method is quite impressive and interesting. I believe it would be potentially beneficial in clinical.  I have only several small issues as shown below:

1)      Figure 1b is not clear.

2)      Some typo errors happened like line 287, what does “platfors” means?

3)      For describing the time point like 2 pm, there should be one blank between the number and the unit.

The quality of English language is fine to me.

Reviewer 3 Report

This manuscript is an interesting paper. It makes a useful contribution to the growing literature on successful strategies to reduce time and cost in pathogen detection through an mNGS workflow using HostEL and AmpRE for both DNA- and RNA-based organisms in a single-reaction sequencing library. The research topic is actual and useful for clinical and biomedical applications. The introduction and conclusion is well written, and all it should be of great interest to the readers.

In my view this is ready for publication but may need some minor changes.

1.    There is no statistical analysis section, and the authors report significant differences in the results.

2.    Are the libraries previously selected before each experiment, or are they inserted once and kept indefinitely? The authors should further clarify the process of introduction and specific selection of the libraries in each experiment.

3.    Can the number of reads per sample vary depending on target? A reliable range should be included showing the number of readings per sample and per time for a typical bacterial, viral and fungal strain.

4.    Is the alignment of the readings with the human genome performed in an automated way?

5.    In figure 3B write in the same format the concentration of bacteria as in figure 3A (for example: 7.0x106/ml).

6.    In Figure 3B it is true that control samples are lost at the higher dilutions, however the readings for other strains are higher in the No HosTEL at these higher dilutions. Please explain in detail the reasons for this.

7.    Figure 3C does not show the significant increases. The authors should include the significant differences in the graph.

8.    Figure 4B is unclear. There are 40 tests detected out of a total of 40? The figure legend should be more detailed.

9.    Figures 1 and 3 have very small texts.

Author Response

1.    There is no statistical analysis section, and the authors report significant differences in the results.
We have updated the data analysis as suggested to include the statical analysis. Briefly, we updated the section to include the following: “Statistical analysis of sensitivity, specificity and clinical agreement was then performed in R. For the analysis, we defined these statistics as: Sensitivity = True positives / (True positives + False negatives) & Specificity = True negatives / (True negatives + False posi-tives); where true positives are the number of spiked-ins that were correctly identified by sequencing as positive. False positives are the number of spiked-ins that were incorrectly identified by the sequencing as present in negative plasma. True negatives are defined as the number of spike-ins that were correctly identified by being not present in negative plasma. False negatives are the number of spike ins that are identified by the sequencing test in negative plasma. For the analysis on clinical agreement, we illustrate the numbers in a cross tabulation illustrating the of occurrence of the different combinations of situations of positive/negative clinical diagnosis vs sequencing positive/negative.”
2.    Are the libraries previously selected before each experiment, or are they inserted once and kept indefinitely? The authors should further clarify the process of introduction and specific selection of the libraries in each experiment.
The sequencing libraries are prepared separately for each condition/patient. We subsequently quantified the molarity of the sequencing libraries using Agilent Tapestation before pooling 5 samples at equimolar for the sequencing run on the Iseq System. We have updated the manuscript in the methods section to include these details. 
3.    Can the number of reads per sample vary depending on target? A reliable range should be included showing the number of readings per sample and per time for a typical bacterial, viral and fungal strain.
    We agree with the reviewer that that the number of reads associated with each pathogen can vary depending on various factors, such as the genomic size, nature of the pathogen towards lysis protocols, the extent of transcriptional activity and hence the number of RNA molecules. Due to the myriad of possible conditions and the limitations of resources, in this study we decided to focus on the digital signal of absent/present by setting a cut-off limit of at least 5 reads assigned to the pathogen to be considered.
4.    Is the alignment of the readings with the human genome performed in an automated way?
We appreciate the reviewer comment that alignment to human genome should be automated. Currently, the alignment is performed in a automated manner using the bioinformatics tool HISAT2. We have updated the manuscript in the data analysis section to reflect the parameters used and reference genome used to perform the alignment.
5.    In figure 3B write in the same format the concentration of bacteria as in figure 3A (for example: 7.0x106/ml).
    We have updated the figure in the manuscript as suggested. 
6.    In Figure 3B it is true that control samples are lost at the higher dilutions, however the readings for other strains are higher in the No HosTEL at these higher dilutions. Please explain in detail the reasons for this.
    In Figure 3B, we are illustrating the diversity of the microbes by showcasing the relative fraction of that each microbe contributed to the non-human reads. Hence when some of the spike-ins are not detected when sequenced, those fraction fall off and are replace by those present. We have since updated the manuscript to clarify this point. 
7.    Figure 3C does not show the significant increases. The authors should include the significant differences in the graph.
    We have included a table in the supplement which illustrates the exact percentage of changes across the different concentrations.
8.    Figure 4B is unclear. There are 40 tests detected out of a total of 40? The figure legend should be more detailed.
    We have updated the figure caption to clarify that there are 4 runs, each with 10 spiked ins for a total of 40 spiked-ins that we expect to detect in the sequencing runs. 
9.    Figures 1 and 3 have very small texts.
    We have updated the fonts size in the manuscript as suggested. 

Reviewer 4 Report

The authors claimed that they developed a metagenomics sequencing workflow for pathogen detection in plasma with low depth. However, it requires more evidence to prove, since the manuscript was not well-written, leaving lots of details out. It is thus difficult for readers to justify their claims.

Major concerns are throughout the manuscript, mostly in Materials and Methods

1)      Lack of quality control and quantification method: NGS is noisy. Without background cut-off clearly defined/described, it makes no sense to claim positive/negative, not even to mention sensitivity or specificity. Besides, each bacterial or fungal genome size is different. NGS is mainly genome-size-based. Without quantification clearly defined/described, it makes no sense to claim how many cells per ml of plasma for sensitivity. Surprisingly, bacteria were used for initial spike-in testing, specimens with viruses were later used for clinical samples.  

2)      Lack of key details: HostEL and AmpRE are difficult to find, though the company name is mentioned. No detail described these products, how much to use, how to use, and so on. While dilution was detailed in sequencing part, 2.2, neither sequencing length nor single- or paired-end was given, not to mention how many reads per sequencing run, how many quality reads in each sequencing run, or quality control for each sample in sequencing. As to data analysis, software versions, kraken2 database source and version are missing, while comprehensiveness of database is key to pathogen detection. For RNA, reverse transcription is missing; for qPCR, PCR for DNA was described, but neither RT-PCR for RNA nor control described.

3)      Lack of precision: “2.1 Analytical validation”: it’s more like initial method testing. “Through specific design”: how specific is it? “Quantifying and identifying pathogen signal”: how to quantify? “Tris-HCl buffered”, pH? “cut-off” for NGS: cannot find the definition. “low-depth sequencing”: how low is the low? “5ml of blood plasma was collected in EDTA tubes”: I believe it’s blood not blood plasma.

Typos or mismatches between context and figures.

Round 2

Reviewer 1 Report

Thank you for your answers. I have no questions about publication.

Author Response

We thank the reviewer for the positive response.

Reviewer 4 Report

A major revision was asked, but receiving a minor revision. Key questions have not been addressed well. As quality control (QC) is essential for clinical diagnosis and NGS is noisy with the bioinformatics analysis method that the authors used, it is critical to clarify in the manuscript. While the authors claimed low depth of NGS required, they only described it as < 1M but not gave the minimum. And I believe the minimum is more important than the low depth as QC. More criteria are needed throughout the manuscript, such as sequencing length, passing filter rate, mapping rate, etc.

The materials and methods part is for readers to repeat the experiments. While the authors described in detail (or too detail) dilutions for the sequencing, they chose to ignore introducing their main methods HostEL and AmpRE. While these products or protocols are unavailable elsewhere, it is difficult for readers to justify. If patented, the patent application numbers should have been given. To me, <2-fold improvement was not impressive, considering the high human host background. 

The NGS methods could be used for virus, bacteria and fungi, as the authors claimed. However, only viral specimens was used for testing, not related at all to their spike-in initial experiments. I wonder how to apply this into clinical diagnosis, if not fully investigated with detailed QC. The sensitivity and specificity are worrisome. If 1000 g 10 min centrifuge is used to remove human host cells, microbial cells might be removed as well, at least reduced. 

Errors persist in the manuscript. For an example, reads per million (TPM), which should be RPM. 

Precision still needed in the manuscript. For an example, database requires collection date as the version.

Please go through the manuscript for the typos.
